# Multitask radiological modality invariant landmark localization using deep reinforcement learning

**Vishwa S. Parekh** [*1,2]                                             VISHWAPAREKH@JHU.EDU
**Alex E. Bocchieri** [*1]                                                    ABOCCHI2@JHU.EDU
**Vladimir Braverman**[1]                                                        VOVA@CS.JHU.EDU
**Michael A. Jacobs** [2,3]                                                   MIKEJ@MRI.JHU.EDU

[1] *Department of Computer Science, The Johns Hopkins University, Baltimore, MD 21218*

[2] *The Russell H. Morgan Department of Radiology and Radiological Sciences, The Johns Hopkins University School of Medicine, Baltimore, MD 21205, USA*

[3] *Sidney Kimmel Comprehensive Center, The Johns Hopkins University School of Medicine, Baltimore, MD 21205, USA*

## Abstract

Deep learning techniques are increasingly being developed for several applications in radiology, for example landmark and organ localization with segmentation. However, these applications to date have been limited in nature, in that, they are restricted to just a single task e.g. localization of tumors or to a specific organ using supervised training by an expert. As a result, to develop a radiological decision support system, it would need to be equipped with potentially hundreds of deep learning models with each model trained for a specific task or organ. This would be both space and computationally expensive. In addition, the true potential of deep learning methods in radiology can only be achieved when the model is adaptable and generalizable to multiple different tasks. To that end, we have developed and implemented a multitask modality invariant deep reinforcement learning framework (MIDRL) for landmark localization and segmentation in radiological applications. MIDRL was evaluated using a diverse data set containing multiparametric MRIs (mpMRI) acquired from different organs and with different imaging parameters. A 2D single agent model was trained to localize six different anatomical structures throughout the body, including, knee, trochanter, heart, kidney, breast nipple, and prostate across T1 weighted, T2 weighted, Dynamic Contrast Enhanced (DCE), Diffusion Weighted Imaging (DWI), and DIXON MRI sequences obtained from twenty-four breast, eight prostate, and twenty five whole body mpMRIs. Additionally, a 3D multi-agent model was trained to localize knee, trochanter, heart, and kidney in the whole body mpMRIs. The trained MIDRL framework produced excellent accuracy in localizing each of the anatomical landmarks. In conclusion, we developed a multitask deep reinforcement learning framework and demonstrated MIDRL's potential towards the development of a general AI for a radiological decision support system.

**Keywords:** multitask, reinforcement learning, landmark, MRI, multiparametric, radiology, deep learning, segmentation

---

* Contributed equally

## 1. Introduction

The field of radiology is moving towards a collaborative space between human experts and artificial intelligence (AI) (Hosny et al., 2018; Parekh and Jacobs, 2019). Automatic anatomical localization is an integral part of an AI radiology framework owing to its diverse applicability across multiple applications such as image segmentation, registration, and classification. For example, in brain image registration, automatic detection of landmarks such as Anterior Commissure (AC) and Posterior Commissure (PC) could be very useful in localizing the mid-sagittal plane (Alansary et al., 2019; Vlontzos et al., 2019). In addition, automatic landmark localization could potentially be used to identify and mark different anatomical landmarks within a computer aided radiological decision support system to create a preliminary draft of results for the radiologists to read. In recent years, deep reinforcement learning (RL) has emerged as one of the best techniques for landmark localization across multiple different research studies (Ghesu et al., 2017; Alansary et al., 2019; Vlontzos et al., 2019). In addition, RL algorithms have been applied in several other radiological applications such as image segmentation, registration, treatment planning, and assessment (Tseng et al., 2017; Ali et al., 2018; Maicas et al., 2017; Ma et al., 2017; Alansary et al., 2018).

Deep RL is an emerging area of active research that has produced excellent results across diverse domains (Mnih et al., 2013, 2015; Li et al., 2016; Silver et al., 2017; Sallab et al., 2017). Briefly, RL deals with an artificial agent that is learning to understand its environment, while attempting to maximize the cumulative award associated with a set of complex tasks (Sutton and Barto, 2018). Every action of the artificial agent is evaluated based on its contribution to the final cumulative reward. In deep RL, deep learning algorithms such as convolutional neural networks (CNN) are used to identify the current state and predict the best possible action. The goal of deep RL algorithms in landmark localization is to learn to locate different anatomical landmarks with high accuracy, computational efficiency, and robustness. However, the applications of deep RL in landmark localization have been restricted, in that, the deep RL is generally trained to localize a landmark (Alansary et al., 2019) or a set of landmarks (Vlontzos et al., 2019) within a predefined anatomical environment (e.g. brain MRI) acquired using specific imaging parameters. Consequently, training multiple deep RLs across different anatomical regions and radiological applications would not only increase the space and time complexity of the application, but would also be difficult to translate in a clinical workflow due the very diverse body regions and diseases. In contrast, applications of deep RL to standard computer vision problems have been more general in scope, for example, a single trained agent playing multiple Atari games (Mnih et al., 2015). Correspondingly, we evaluated the feasibility of training a multitask RL agent that could detect anatomical landmarks across different anatomical environments and imaging parameters, as illustrated in Figure 1. In this work, we developed and implemented a multitask modality invariant deep reinforcement learning (MIDRL) framework and evaluated it to localize six different anatomical structures throughout the body, including, the chest (heart,breast), abdomen (kidney,prostate), pelvis (lesser trochanter) and lower extremity (knee) using T1-weighted (T1WI), T2-weighted (T2WI), Dynamic Contrast Enhanced (DCE), Diffusion Weighted Imaging (DWI) with Apparent

Diffusion Coefficient (ADC) mapping, and DIXON MRI sequences obtained from multiparametric breast, prostate, and whole body (WB) MRI acquisitions.

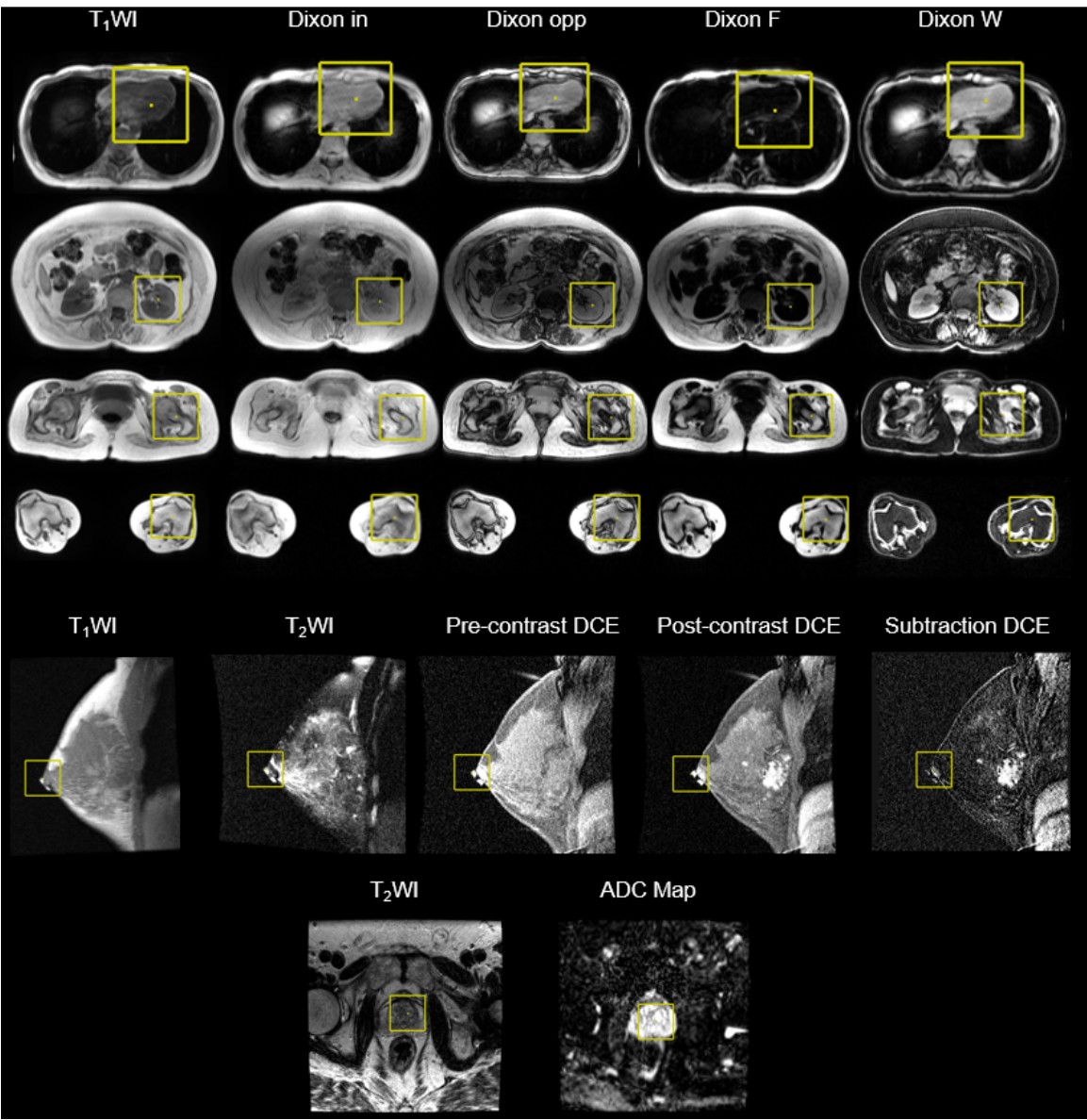

Figure 1: Illustration of a deep RL agent trained to localize different anatomical landmarks on a diverse multi-organ dataset with different imaging parameters of T1-weighted (T1WI), T2-weighted(T2WI), Dynamic Contrast Enhanced (DCE), Diffusion Weighted Imaging (DWI) with Apparent Diffusion Coefficient (ADC) mapping and DIXON

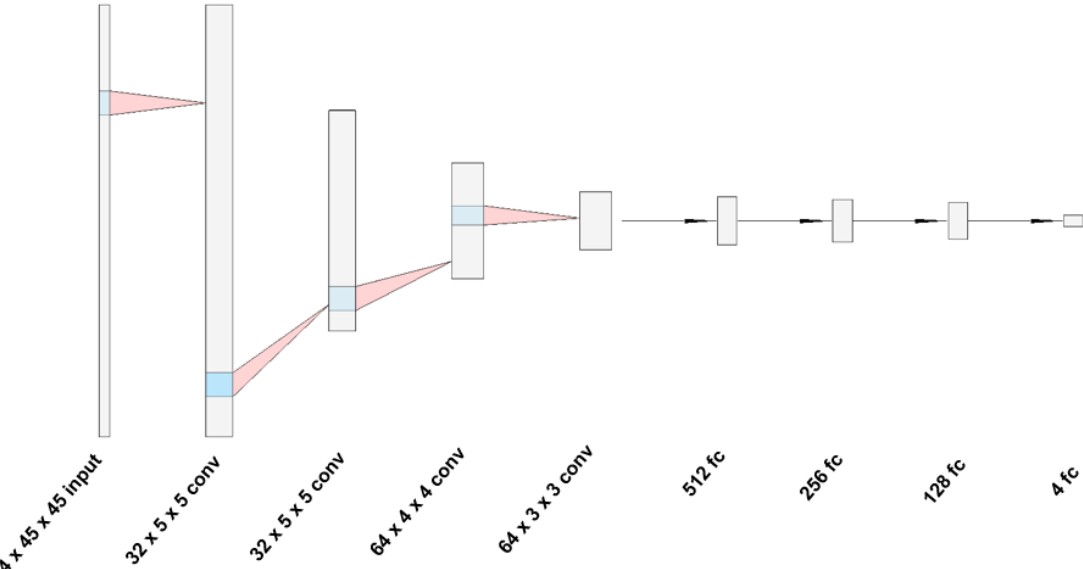

Figure 2: The 2D DQN architecture. There is 2x2 maxpooling between each convolutional layer. The last fully connected layer is of size 4 corresponding to the possible actions: move up, down, left, or right.

## 2. Methods

### 2.1. Clinical data

All studies were performed in accordance with the institutional guidelines for clinical research under a protocol approved by our Institutional Review Board (IRB) and all HIPAA agreements were followed for this retrospective study. The clinical data consisted of 57 mpMRIs. These datasets included 25 whole body (44%), 24 (42%) breast and eight (14%) prostate mpMRIs. The patient population and image acquisition parameters have been detailed in the following subsections.

#### 2.1.1. BREAST MPMRI

The breast mpMRI dataset consisted of 12 women with malignant lesions imaged at 1.5T at two timepoints for developing treatment response metrics. The mpMRI was acquired before and after the first cycle of chemotherapy. The imaging protocol included sagittal fat suppressed (FS) T2WI spin echo (TR/TE=5700/102ms) and fast spoiled gradient echo images (FSPGR) T1WI (TR/TE = 200/4.4 ms) with field of view (FOV)=18cmx18cm, matrix=256x192, slice thickness (ST): 4 mm, 1mm gap). Dynamic contrast enhanced (DCE) FSPGR T1WI (TR/TE=20/4ms, matrix=512x160, ST: 2 mm, 3-4 phases after injection) was obtained after intravenous administration of GdDTPA contrast agent (Omniscan, Amersham Health, 0.2 mL/kg (0.1 mmol/kg)). The contrast agent was injected over 10 seconds with MR imaging beginning immediately after completion of the injection. The

contrast bolus was followed by a 20cc saline flush. Total scan time was less than 20 minutes. In summary, five different images – T1WI, T2WI, Pre-contrast DCE, Post-contrast DCE, and subtraction DCE were evaluated for the detection of nipple within the breast images.

### 2.1.2. Prostate mpMRI

The prostate mpMRI data set consisted of 8 patients with prostate cancer. The mpMRI parameters were were T2WI(TR/TE-3000/30, FOV=240cmx240cm, Matrix=256x256, Slice thickness(ST)=3mm, NEX=2), Trace DWI (TR/TE=2000-3000/70-42, FOV=221x250, Matrix=256x256, ST=3mm, b-values = 0, 400, 800). ADC maps were constructed using a monoexponential equation. The imaging parameters of T2WI and ADC map were evaluated in this study for localization of prostate within the images.

### 2.1.3. Whole body mpMRI

The WB mpMRI data set consisted of 25 subjects acquired using the imaging protocol that scanned from the shoulders to the lower mid calf and described in (Leung et al., 2020). Of the 25 subjects, there were 19 patients with muscular dystrophy and 6 normal volunteers. The imaging parameters of T1WI, T2WI and DIXON weighted images were acquired at 3T and evaluated in this study for detection of heart, left kidney, left trochanter, and left knee cap.

### 2.2. Multitask modality invariant deep reinforcement learning framework

This paper investigates landmark localization carried out by an artificial agent trained using deep Q-learning adapted from (Alansary et al., 2019) and (Vlontzos et al., 2019). (Mnih et al., 2015) originally proposed training deep Q-networks (DQN) for approximating Q-functions. (Alansary et al., 2019) investigated various DQN architectures, including standard DQN(Mnih et al., 2015), Double DQN (Hasselt, 2010; Van Hasselt et al., 2016) and Duel DQN (Wang et al., 2015), for landmark localization in 3D images for a single task. (Vlontzos et al., 2019) used multi-agent DQNs for locating multiple landmarks within single anatomical environments. In this work, we train a 2D single agent DQN and a 3D multi-agent DQN for locating different landmarks using different imaging parameters acquired from multiple anatomical environments (e.g. breast MRI, prostate MRI and whole body MRI). The implementation details for the MIDRL framework will be detailed in the following subsections.

### 2.3. Deep Q-Learning

The Q-learning algorithm attempts to learn a policy for maximizing the expected total future reward. Policy $\pi$ dictates what action $a$ the artificial agent takes in state $s$. The DQN approximates the optimal state-action value function (Q-function)

$$Q(s, a) = \max_\pi E\big[r_t + \gamma r_{t+1} + \gamma^2 r_{t+2} + \ldots | s_t = s, a_t = a, \pi\big] \tag{1}$$

where $r_t$ is the reward at time step $t$ and $\gamma \in [0, 1]$ is the discount factor. Using the Bellman equation, $Q(s, a)$ can be solved iteratively in the form of

$$Q_{i+1}(s, a) = E\big[r + \gamma \max_{a'} Q_i(s', a')\big] \tag{2}$$

where $i$ is the iteration, $s'$ is the next state, and $a'$ is the next action.

In the 2D case, the input to the DQN is a state $s$, which is a sequence of areas (frames) of the 2D image cropped by the agent's bounding box. More specifically, one input sample with a sequence of length $C$ frames has dimensions $C \times H \times W$, where $C$ is the number of channels and $H \times W$ are the dimensions of the cropped areas. The agent's performance can be improved by adopting the multiscale technique, in which, cropped areas are obtained by sampling the 2D image with a stride. For example, a multiscale state is always $H \times W$ in size, but it can be sampled from a larger area of the image. The sampling stride is initially large to cover a larger area of the image at a lower resolution. When the agent converges (oscillates around a point), both the sampling stride and action step size decrease, and the agent sees a smaller area of the image at a higher resolution. Changing the agent's field of view and step size in this way results in faster convergence to the target landmark. The 3D case follows by including depth $D$ in the dimensions of the cropped area.

The training process uses a target network, $DQN_T$, and a policy network, $DQN_P$, both with the same architecture. Upon initialization, both networks have the same weights. During one training step, the input to $DQN_P$ is state $s$, and the input to $DQN_T$ is the next state $s'$. $DQN_P$ outputs the predicted Q-value $Q^*(s, a)$, and $DQN_T$ outputs $Q^*(s', a')$. The target Q-value is $r + \gamma Q^*(s', a')$. After computing the loss function, $DQN_P$'s weights are updated on each step using stochastic gradient descent. $DQN_T$'s weights are only updated to $DQN_P$'s weights every $N$ steps. This is done to stabilize training.

Training samples are selected from an experience replay buffer. During training, an experience replay buffer is populated with the state, action, reward, and resulting state $[s, a, r, s']$ from steps taken over many episodes. Steps are taken according to an $\epsilon$-greedy policy, where an action is taken uniformly at random with probability $\epsilon$ at each step. Otherwise, the action with the highest reward is chosen.

Overall, the reinforcement learning framework is composed as follows. The environment is an image in which the agent is a bounding box that has four possible actions in a 2D slice or six possible actions in a 3D volume. Actions include $a \in \{$x++, x--, y++, y--, z++, z--$\}$. The state is the sequence of cropped areas as previously described above. The reward is the difference between the Euclidean distance from the target landmark to the agent before and after the agent takes a step in a certain direction. The reward is positive if the agent moves closer to the landmark and negative if it moves farther. The landmark's location is its $(x, y, z)$ coordinate in the image, and the agent's location is the bounding box's center $(x, y, z)$ coordinate.

## 2.4. Training Details

The DQN is composed of four 2D convolutional layers with maxpooling and PReLU activations and four fully connected layers with leaky ReLU activations. See Figure 2.

Stochastic gradient descent is performed using the Adam optimizer with learning rate $= 0.001$, $\beta_1 = 0.9$, $\beta_2 = 0.999$, $\epsilon = 0.001$. The Huber loss function is used. Training samples have a frame history of length $C = 4$, cropping of size $H \times W = 45 \times 45$, and batch size of 48. Discount factor $\gamma = 0.9$. Reward $r$ is clipped to between -1 and 1. $\epsilon = 1$ at epoch 1 decreases to $\epsilon = 0.1$ by epoch 10. Training lasts for 20 epochs with 25,000 steps per

epoch. The target DQN is updated every $N = 2,500$ steps. The agent follows the multiscale technique with sampling strides 3, 2, 1 corresponding with action step sizes 9, 3, 1.

The complete MIDRL framework was evaluated by dividing the data set into train and test sets with a 70-30 split. Consequently, the training set consisted of 17 whole body mpMRIs, 9 breast mpMRIs, and 5 prostate mpMRIs, and the test set consisted of 8 whole body mpMRIs, 3 breast mpMRIs, and 3 prostate mpMRIs. The resultant anatomical localization on the test set were evaluated by computing the error in terms of distance between the target and agent's location. The Dice Similarity (DS) between the terminal agent's and ground truth bounding boxes from each landmark are calculated.

We also evaluated an analogous 3D multi-agent model adapted from (Vlontzos et al., 2019) on 3D whole body mpMRIs. The model was trained to locate the same heart, kidney, trochanter, and knee landmarks from the 2D single agent experiment, now within their original 3D whole body image. This model's network is composed of the same layers as the 2D model but with their 3D counterparts. The convolutional layers of the network are shared by all agents, but each agent has its own individual set of final fully connected layers. The model has four agents with bounding boxes of size $45 \times 45 \times 11$ that encompass the 3D anatomical objects. Each agent is assigned to locate one landmark. The 3D mutli-agent model was tested on the same whole body imaging parameters (Dixon in phase, Dixon out of phase, Dixon fat, Dixon water, T1WI, T2WI) as in the 2D single agent experiment. Reported results include the 3D overlap between agent and target bounding boxes and the 3D distance between the bounding boxes' centers. Figure 3 shows an example of the multi-agent model locating landmarks.

Source code is publicly available for the 2D single agent [1] and 3D multi-agent [2] models.

## 3. Results

A total of 57 mpMRI data sets were evaluated for anatomical localization of breast nipple, prostate, left kidney, left trochanter, left knee cap, and heart using a diverse set of anatomical locations and mpMRI parameters consisting of T1WI, T2WI, DCE-MRI, DIXON, and DWI. Each of the patient subsets were approximately divided into train/test sets with 70% used for training and 30% for testing.

For the 2D single agent experiment, Table 1 summarizes IoU, and Table 2 summarizes the distance error (in mm) for each of the subsets. The MIDRL framework performed best in localizing the knee with an average IoU of 0.86 and average distance error of 4.5 mm. The worst performance was observed in breast with an average IoU of 0.47 and average distance error of 16.9 mm. The nipple localization failed in most of the post contrast and subtraction contrast images, where, the tumor was incorrectly localized as the nipple.

For the 3D multi-agent experiment, Table 3 summarizes IoU and Table 4 summarizes the distance error (in mm) for locating the heart, left kidney, left trochanter, and left knee in 3D whole body MRI. The best performance was observed in localizing the trochanter with an average IoU of 0.70 and an average distance error of 10.4 mm. The worst performance was observed in localizing the knee with an average IoU of 0.49 and an average distance error of 52.5 mm. At times, the model located the bone in the thigh instead of the knee.

---

1. 2D single agent source code
2. 3D multi-agent source code

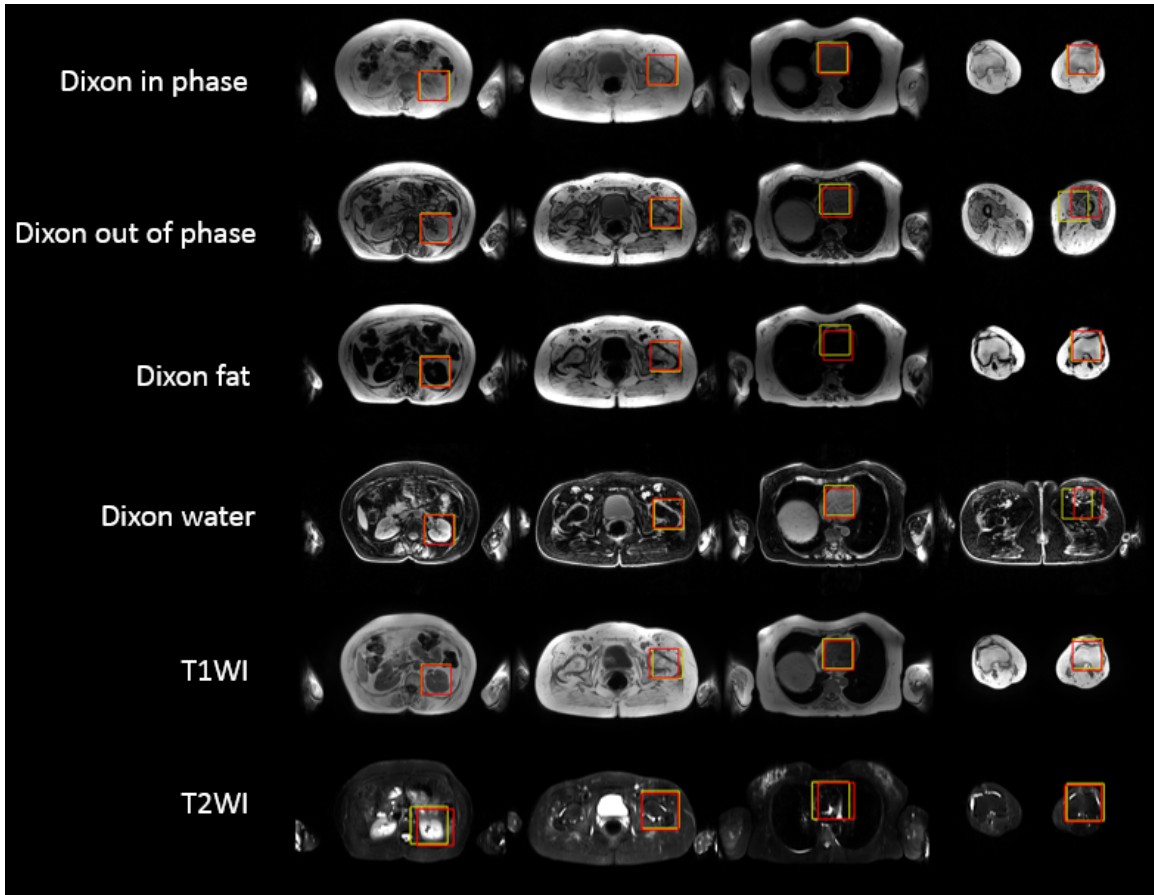

Figure 3: An example of the 3D multi-agent model locating landmarks in Dixon in phase, Dixon out of phase, Dixon fat, Dixon water, T1WI, and T2WI whole body images of a patient. Each agent locates one landmark. Landmarks from left to right are the left kidney, left trochanter, heart, and left knee. The agent's bounding box is yellow, and the target's bounding box is red. In this example, the model failed to locate the knee slice in the Dixon out of phase image and the Dixon water image.

## 4. Discussion

We demonstrated the feasibility of training deep RL agents for anatomical localization across a diverse set of anatomical environments and imaging parameters with excellent results. These results assert the possibility of a general AI framework based on deep RL for anatomical landmark localization in radiological applications.

Deep reinforcement learning has been successfully applied for localization of single anatomical landmarks in radiological imaging. Recently, multi-agent deep reinforcement learning with collaborating agents were evaluated in the radiological setting for detection of multiple landmarks in brain MRI, cardiac MRI, and fetal brain ultrasound with excellent results (Vlontzos et al., 2019). However, these models were limited to a single anatomical environment and not in a multi-environmental model. In contrast, we successfully trained a single deep RL agent to localize different anatomical landmarks across diverse data sets that were acquired with different fields of view, magnetic field strengths (1.5T and 3T), imaging orientations (sagittal and axial), and imaging parameters (T1WI, T2WI, DCE, DWI, DIXON), demonstrating the robustness of the MIDRL framework.

Training a deep reinforcement learning model is a computationally expensive process (Alansary et al., 2019) making individual landmark detection impractical for clinical translation. In this work, we explored the multitask capability of deep RL agents (MIDRL) in a diverse environment with excellent success, thereby, improving upon both the space and time complexity of the existing deep RL frameworks.

Our study had certain limitations. The deep RL agent failed to identify nipple in most of the post-contrast and subtract DCE images. We observed that the tumor was incorrectly identified on breast images as nipple in the cases the deep RL agent failed. One possible reason for this could be the fact that the spiculations in the tumor were incorrectly identified due to their shape as the neighborhood around the nipple. We are currently investigating this to correctly identify the cause of incorrect nipple localization and improve efficacy of the MIDRL framework. MIDRL also performed poorly on the T2WI trochanter possibly due to the image's proximity and similarity to the prostate.

The deep RL framework in its current implementation is not equipped to return NULL when a target is not found. In addition, a slice may contain multiple landmarks of interest as opposed to just one. In conclusion, the MIDRL framework initiates a first step towards a universal deep reinforcement learning framework capable of identifying and localizing anatomical landmarks irrespective of the imaging modality, underlying anatomical environment, and image acquisition parameters.

Table 1: 2D IoU (mean ± stdev) for the 2D single agent experiment. IoU is between the agent's terminal bounding box and the bounding box centered on the target landmark. Both bounding boxes are of size $45 \times 45$. A missing entry indicates that the dataset does not include that imaging parameter for that landmark.

2D Single Agent IoU

|  | Heart | Kidney | Trochanter | Knee | Breast | Prostate |
|---|---|---|---|---|---|---|
| T1WI | 0.75 ± 0.20 | 0.70 ± 0.35 | 0.74 ± 0.23 | 0.88 ± 0.06 | 0.78 ± 0.16 | |
| T2WI | 0.58 ± 0.21 | 0.59 ± 0.20 | 0.29 ± 0.36 | 0.68 ± 0.11 | 0.51 ± 0.29 | 0.48 ± 0.42 |
| Dixon in | 0.79 ± 0.12 | 0.62 ± 0.25 | 0.85 ± 0.06 | 0.88 ± 0.05 | | |
| Dixon opp | 0.77 ± 0.27 | 0.74 ± 0.17 | 0.83 ± 0.09 | 0.87 ± 0.08 | | |
| Dixon F | 0.78 ± 0.11 | 0.69 ± 0.33 | 0.78 ± 0.14 | 0.87 ± 0.03 | | |
| Dixon W | 0.82 ± 0.14 | 0.70 ± 0.35 | 0.75 ± 0.18 | 0.91 ± 0.03 | | |
| Post DCE | | | | | 0.20 ± 0.32 | |
| Pre DCE | | | | | 0.50 ± 0.42 | |
| Sub DCE | | | | | 0.12 ± 0.24 | |
| ADC | | | | | | 0.34 ± 0.36 |
| All parameters | 0.76 ± 0.19 | 0.68 ± 0.28 | 0.73 ± 0.25 | 0.86 ± 0.09 | 0.47 ± 0.37 | 0.41 ± 0.36 |

Table 2: 2D distance errors in mm (mean ± stdev) for the 2D single agent experiment. The distance between the agent's terminal location (center of its bounding box) and the target landmark. A missing entry indicates that the dataset does not include that imaging parameter for that landmark.

2D Single Agent Distance Errors (mm)

|  | Heart | Kidney | Trochanter | Knee | Breast | Prostate |
|---|---|---|---|---|---|---|
| T1WI | 9.8 ± 10.4 | 23.1 ± 46.2 | 9.6 ± 10.2 | 3.5 ± 2.4 | 3.1 ± 3.4 | |
| T2WI | 22.3 ± 17.7 | 19.0 ± 12.7 | 68.2 ± 51.8 | 13.4 ± 5.3 | 9.5 ± 7.4 | 20.6 ± 21.2 |
| Dixon in | 7.0 ± 4.3 | 15.1 ± 13.8 | 5.5 ± 2.2 | 3.6 ± 1.2 | | |
| Dixon opp | 10.1 ± 16.8 | 11.6 ± 12.1 | 6.4 ± 3.2 | 3.8 ± 2.7 | | |
| Dixon F | 7.4 ± 4.8 | 17.7 ± 30.0 | 7.9 ± 4.7 | 3.8 ± 0.7 | | |
| Dixon W | 6.2 ± 5.6 | 15.6 ± 24.6 | 9.8 ± 8.5 | 2.3 ± 0.7 | | |
| Post DCE | | | | | 30.0 ± 29.5 | |
| Pre DCE | | | | | 3.3 ± 2.8 | |
| Sub DCE | | | | | 38.6 ± 28.7 | |
| ADC | | | | | | 16.7 ± 24.8 |
| All parameters | 10.0 ± 11.5 | 16.9 ± 25.5 | 15.7 ± 27.5 | 4.5 ± 3.9 | 16.9 ± 22.9 | 18.7 ± 20.7 |

Table 3: 3D IoU (mean ± stdev) for the 3D whole body multi-agent experiment. IoU is between the multi-agent's terminal bounding box and the bounding box centered on the target landmark. Bounding boxes are of size $45 \times 45 \times 11$.

3D Multi-Agent IoU

|  | **Heart** | **Kidney** | **Trochanter** | **Knee** |
|---|---|---|---|---|
| T1WI | 0.62 ± 0.21 | 0.66 ± 0.19 | 0.78 ± 0.10 | 0.43 ± 0.34 |
| T2WI | 0.47 ± 0.14 | 0.44 ± 0.32 | 0.34 ± 0.26 | 0.35 ± 0.37 |
| Dixon in | 0.71 ± 0.14 | 0.54 ± 0.23 | 0.81 ± 0.10 | 0.45 ± 0.39 |
| Dixon opp | 0.62 ± 0.19 | 0.71 ± 0.14 | 0.74 ± 0.13 | 0.51 ± 0.42 |
| Dixon F | 0.48 ± 0.29 | 0.51 ± 0.29 | 0.85 ± 0.10 | 0.69 ± 0.29 |
| Dixon W | 0.65 ± 0.13 | 0.75 ± 0.14 | 0.60 ± 0.25 | 0.49 ± 0.41 |
| All parameters | 0.60 ± 0.20 | 0.61 ± 0.24 | 0.70 ± 0.23 | 0.49 ± 0.37 |

Table 4: 3D distance errors in mm (mean ± stdev) for the 3D whole body multi-agent experiment. The distance between the agent's terminal location (center of its bounding box) and the target landmark.

3D Multi-Agent Distance Errors (mm)

|  | **Heart** | **Kidney** | **Trochanter** | **Knee** |
|---|---|---|---|---|
| T1WI | 12.6 ± 9.4 | 10.2 ± 6.5 | 6.4 ± 2.7 | 22.9 ± 20.4 |
| T2WI | 18.0 ± 7.6 | 29.3 ± 33.2 | 27.6 ± 17.1 | 91.7 ± 126.2 |
| Dixon in | 8.1 ± 4.7 | 18.3 ± 21.8 | 5.5 ± 2.6 | 51.6 ± 64.0 |
| Dixon opp | 11.4 ± 6.5 | 8.5 ± 4.5 | 7.0 ± 3.2 | 57.4 ± 78.5 |
| Dixon F | 21.7 ± 17.9 | 39.2 ± 72.0 | 4.2 ± 2.4 | 32.3 ± 75.9 |
| Dixon W | 10.1 ± 4.8 | 6.9 ± 4.0 | 15.9 ± 14.5 | 68.7 ± 92.6 |
| All parameters | 13.5 ± 10.3 | 18.3 ± 34.1 | 10.4 ± 11.4 | 52.5 ± 78.4 |

## Acknowledgments

Funding: This work was supported by the National Institutes of Health grant numbers: 5P30CA006973 (IRAT), U01CA140204, and 1R01CA19029.

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
