# OpenReview forum: "Multitask  radiological modality invariant landmark localization using deep reinforcement learning"
_MIDL.io/2020/Conference — MIDL 2020_

### Official Review · AnonReviewer1 · 2020-02-23
**interesting RL agent environment with overstated contribution**

**Rating:** 3
**Confidence:** 5
**Recommendation:** Poster

**Summary:**

This paper proposed to use a single DQN agent to localise different landmarks in different MRI imaging sequences. An evaluation is provided for localizing six different anatomical structures throughout the body, including, knee, trochanter, heart, kidney, breast nipple, and prostate across T1 weighted, T2 weighted, Dynamic Contrast Enhanced (DCE), Diffusion Weighted Imaging (DWI), and DIXON MRI sequences obtained from twenty-four breast, eight prostate, and twenty five whole body mpMRIs.

**Strengths:**

- RL is promising for assistance tasks
- the angle of this work is interesting: Train a single agent for many modality-dependant tasks. This hasn't been done previously.
- the data-set and resulting training environment is interesting

**Weaknesses:**

- the authors claim to introduce the 'MIDRL' framework, which implies novelty. However, the used framework is a DQN with various training modalities. No special framework is introduced, that would multi-modality environments make possible.
- The task and modality are dependant. The CNN might just use it's capacity to propose actions for each modality independently. Some overlap might exist for the tasks because the same organ is targeted in several multi-modal acquisitions. What would happen if different target landmarks in the same modality would define the task?
- This is a 2D method that will always return a localisation for each slice as stated in the paper's discussion. "The deep RL
framework in its current implementation is not equipped to return NULL when a target is not found.". How are all the slices handled during the evaluation that don't contain the landmark? Is this only evaluated on slices that are guaranteed to contain the landmark?
- what hasn't a 3D method been used as common in the frequently cited related work?
- It's not quite clear why are there results missing in the tables?
- the contribution is overstated. The authors' argument is that previous 'models were limited to a single anatomical environment' and that their method is the first that provides 'a multi-environmental model'. As soon as several landmarks need to be found in each of the modalities, this approach breaks down. First, as shown by Alansary et al., different landmarks require different training and DQN strategies and as shown by Vlontzos et al., multiple landmarks perform best with multiple agents that share weights. Testing Vlontzos' work in the proposed multi-modality environment for multiple landmarks per modality would be interesting, but this hasn't been done in the manuscript.

**Detailed Comments:**

Please see above. Overall the paper is interesting but without methodology novelty. The problem statement is the original part of this work. A through investigation of existing methods in the newly proposed setup would have been better than to try and introduce novelty through claiming a new 'framework'.

**Justification Of Rating:**

The paper is ok, but would probably be better accepted as an abstract with poster. RL is clearly an interesting area for medical image analysis and should be discussed at MIDL, however the paper should not be published as full paper in its current form.


**Paper Type:**

validation/application paper

**Questions To Address In The Rebuttal:**

- how would Alansary et al. and Vlontzos et al 3D methods perform for this setup. Their code is publicly available on github.
- will the data used in the paper be made publicly available, same for the code.
- can you comment on my questions above re evaluation on slices that do no contain a landmark please?
- why are there results missing in the tables?

**Special Issue:**

no

---

> ### Author Response · Authors · 2020-03-28
> **Response to Reviewer 1**
>
> 1. "how would Alansary et al. and Vlontzos et al 3D methods perform for this setup. Their code is publicly available on github.”
>
> Performance would depend on the 3D volume's field of view. If one 3D volume contains multiple landmarks (for example, a whole body MRI with knee, trochanter, kidney, and heart as landmarks) then a single-agent model (Alansary et al.) would likely locate the landmark closest to the agent's initial location. A multi-agent model (Vlontzos et al.) should be able to locate all landmarks (one agent per landmark).
>
> Following the reviewers’ remarks, we modified and implemented Vlontzos et al.’s multi-agent code on our 3D whole body Dixon, T1, T2 images to locate the same four landmarks we originally describe in our paper (knee, trochanter, kidney, and heart). The single multi-agent model was able to locate each of the four landmarks in the 3D volume for each of the imaging parameters: Dixon in phase, Dixon out of phase, Dixon fat, Dixon water, T1WI, T2WI. This reinforces our findings that one model can learn multiple landmarks across different environments (imaging parameters). We would like to refer Reviewer 1 to our response to Reviewer 4 where we have summarized results from this experiment. We will add the multi-agent experimental details and results in our revised manuscript.
>
> 2. "will the data used in the paper be made publicly available, same for the code."
>
> Our code would be hosted on github and freely available to everyone. All relevant clinical data will be available upon request with adherence to HIPPA laws and the institution’s IRB policies.
>
> 3. "can you comment on my questions above re evaluation on slices that do no contain a landmark please?"
>
> The reviewer has raised a very important point. In this preliminary work, we have only considered slices/volumes that contain a desired landmark. Models are trained and evaluated on slices that are guaranteed to have a landmark. We are currently working on updating the framework such that it returns NULL when the desired landmark is not found.
>
>  4. "why are there results missing in the tables?"
>
> A missing entry means we did not have that imaging parameter for that landmark in our dataset (N/A). For example, the heart, kidney, trochanter, and knee images we evaluated did not have Post DCE, Pre DCE, Sub DCE, or ADC images in their respective multiparametric MRI dataset.

---

> > ### Comment · AnonReviewer1 · 2020-03-29
> > **excellent reply. I would change my recommendation to strong accept.**
> >
> > could you please add this information to the paper or an appendix?

---

> > > ### Author Response · Authors · 2020-04-03
> > > **Revised manuscript**
> > >
> > > We have revised the manuscript to include the 3D multiagent experiment and results. However, we cannot find how to upload the revision?

---

### Official Review · AnonReviewer3 · 2020-03-09
**Multitask radiological modality invariant landmark localization using deep reinforcement learning**

**Rating:** 2
**Confidence:** 4
**Recommendation:** Poster

**Summary:**

The aim of this paper is to develop and implement a multitask modality invariant deep reinforcement learning for landmark localization and segmentation in radiological application. Topic is interesting. However, there are several concerns on this paper.

The strength of this paper.
1. Good topics to implement a multitask modality invariant deep reinforcement learning for landmark localization and segmentation in radiological application

The weakness of this paper.
1. In case of bounding box, please use intersection of union (IoU) metric.
2. There is a lack of ablation studies
3. What's the FROC evaluation on this detection issues? In test set, an image don't have interesting related regions necessarily.
4. In table 2, the unit of distance error is pixels. However, pixel number depends on the FOV of images. Therefore, I recommend to use absolute mm as the unit.
5. small number of samples in training and test dataset. In addition, there is no external or cross-validations.

**Strengths:**

Good topics
to implement a multitask modality invariant deep reinforcement learning for landmark localization and segmentation in radiological application
to show feasibility of training a single deep RL agent for multitask modality invariant applications.


**Weaknesses:**

1. In case of bounding box, please use intersection of union (IoU) metric.
2. There is a lack of ablation studies
3. What's the FROC evaluation on this detection issues? In test set, an image don't have interesting related regions necessarily.
4. In table 2, the unit of distance error is pixels. However, pixel number depends on the FOV of images. Therefore, I recommend to use absolute mm as the unit.
5. small number of samples in training and test dataset. In addition, there is no external or cross-validations.

**Detailed Comments:**

...

**Justification Of Rating:**

there are several concerns on this paper.

1. In case of bounding box, please use intersection of union (IoU) metric.
2. There is a lack of ablation studies
3. What's the FROC evaluation on this detection issues? In test set, an image don't have interesting related regions necessarily.
4. In table 2, the unit of distance error is pixels. However, pixel number depends on the FOV of images. Therefore, I recommend to use absolute mm as the unit.
5. small number of samples in training and test dataset. In addition, there is no external or cross-validations

**Paper Type:**

both

**Questions To Address In The Rebuttal:**

1. In case of bounding box, please use intersection of union (IoU) metric.
2. There is a lack of ablation studies
3. What's the FROC evaluation on this detection issues? In test set, an image don't have interesting related regions necessarily.
4. In table 2, the unit of distance error is pixels. However, pixel number depends on the FOV of images. Therefore, I recommend to use absolute mm as the unit.
5. small number of samples in training and test dataset. In addition, there is no external or cross-validations.

**Special Issue:**

no

---

> ### Author Response · Authors · 2020-03-28
> **Response to Reviewer 3**
>
> 1. In case of bounding box, please use intersection of union (IoU) metric.
>
> We thank the reviewer for their suggestion, and have now updated the results with the IoU metric. The results with the IoU metric over all corresponding imaging parameters have been summarized below (avg $\pm$ stdev) :
>
> Heart: 0.76 $\pm$ 0.19
> Kidney: 0.68 $\pm$ 0.28
> Trochanter: 0.73 $\pm$ 0.25
> Knee: 0.86 $\pm$ 0.09
> Breast: 0.47 $\pm$ 0.37
> Prostate: 0.41 $\pm$ 0.36
>
> 2. There is a lack of ablation studies
>
> We thank the reviewer for their suggestion. In response to your remark, we have now started working on the ablation studies. However, due to lack of time (1 week), we do not have any results available on the ablation studies.
>
> 3. What's the FROC evaluation on this detection issues? In test set, an image don't have interesting related regions necessarily.
>
> The reviewer has raised a very important point. In this preliminary work, we have only considered slices/volumes that contain a desired landmark. Models are trained and evaluated on slices that are guaranteed to have a landmark. We are currently working on updating the framework such that it returns NULL when the desired landmark is not found.
>
> 4. In table 2, the unit of distance error is pixels. However, pixel number depends on the FOV of images. Therefore, I recommend to use absolute mm as the unit.
>
> We thank the reviewer for their suggestion. We have updated the tables with absolute mm as the unit. The results with the absolute mm over all corresponding imaging parameters have been summarized below (avg $\pm$ stdev):
>
> Heart: 10.0 $\pm$ 11.5
> Kidney: 16.9 $\pm$ 25.5
> Trochanter: 15.7 $\pm$ 27.5
> Knee: 4.5 $\pm$ 3.9
> Breast: 16.9 $\pm$ 22.9
> Prostate: 18.7 $\pm$ 20.7
>
> 5. small number of samples in training and test dataset. In addition, there is no external or cross-validations
>
>  In this manuscript, our focus is on demonstrating the proof of concept that a single model could be used to identify landmarks across multiple imaging modalities. In the future, we plan to extensively validate this approach using a large dataset with diverse landmarks and imaging modalities.

---

### Official Review · AnonReviewer4 · 2020-03-14
**Interesting approach to landmark detection; some strange examples**

**Rating:** 3
**Confidence:** 4
**Recommendation:** Poster

**Summary:**

The authors present a reinforcement learning approach to landmark detection. They show that the same model can perform several different tasks (nipple detection, prostate detection and organ detection in MRI) and show the effectiveness. They argue that such an approach is better than having different models for different tasks.

The tasks presented are not that novel, and neither is the method, but it is an interesting result nevertheless and can be used for other researchers to be expanded upon.

**Strengths:**

- It is interesting to see RL applied to landmark detection in medical imaging
- The authors show the model performs well on several different applications and MRI sequences
- The evaluation is bit limited, but appears sound

**Weaknesses:**

- I see no use for nipple detection in MRI. I can imagine that this works as there typically is only one nipple, while you can have several lesions. I would like to see argued better why this is relevant.
- There is no comparison with other methods to detect these organs. While I see that they likely are more task specific, setting everything up well should allow to retrain the models for a different task as well
- The method is purely 2D, which is nowadays uncommon for medical images.

**Justification Of Rating:**

- Application paper. Neither method nor task is new.
- It is an important question in medical imaging.
- Experiments have been well thought, but are sometimes lacking in relevance.
- Methodology appears sound.

**Paper Type:**

validation/application paper

**Questions To Address In The Rebuttal:**

Please argue against the weaknesses I proposed. It will be hard to change my mind, as I do not think it warrants a strong accept, unless you are able to completely expand the experiments.

**Special Issue:**

no

---

> ### Author Response · Authors · 2020-03-28
> **Response to Reviewer 4**
>
> 1. "I see no use for nipple detection in MRI. I can imagine that this works as there typically is only one nipple, while you can have several lesions. I would like to see argued better why this is relevant."
>
> The nipple is used as a reference point in many applications. BI-RADS uses the nipple as a reference point for locating a lesion. The nipple is a landmark commonly used in breast image registration. Computer aided detection algorithms often use the nipple as the coordinate system's origin in a mammogram. Our breast dataset was composed of MR images, but our work suggests that one model should perform well when including mammograms in the dataset as well.
>
> 2. "There is no comparison with other methods to detect these organs. While I see that they likely are more task specific, setting everything up well should allow to retrain the models for a different task as well"
>
> The reviewer raises an interesting point. However, the whole idea of this manuscript is to avoid re-training task-specific models for different tasks. In contrast, the idea of this manuscript is to train a single multi-task model that can perform multiple tasks.
>
>  3. "The method is purely 2D, which is nowadays uncommon for medical images."
>
> We thank the reviewer for their suggestion. We started with a 2D implementation to demonstrate proof of concept. Following this suggestion, we have now extended our work and trained a multi-agent model to locate different landmarks in a 3D whole body volume across different imaging parameters. We will add the experimental details and the corresponding results to the revised manuscript.
>
> The experiment was performed on 3D whole body volumes from Dixon in, Dixon opp, Dixon F, Dixon W, T1WI, and T2WI imaging parameters. The summary of results from this experiment is as follows:
>
> Distance error (avg $\pm$ stdev) in absolute mm over all imaging parameters:
>
> Heart: 30.4 $\pm$ 44.8
> Kidney: 24.1 $\pm$ 42.3
> Trochanter: 17.4 $\pm$ 43.9
> Knee: 42.1 $\pm$ 62.6
>
> IoU between agent and target 2D bounding boxes (avg $\pm$ stdev) over all imaging parameters:
>
> Heart: 0.74 $\pm$ 0.11
> Kidney: 0.79 $\pm$ 0.16
> Trochanter: 0.80 $\pm$ 0.14
> Knee: 0.74 $\pm$ 0.23

---

### Meta-Review · Area_Chair1 · 2020-04-07
**MetaReview of Paper273 by AreaChair1**

**Rating:** 3
**Recommendation For Accepted Papers:** Poster

**Metareview:**

An interesting application of reinforcement learning in medical imaging. The reviewers seem to agree on this as well. Two out of three reviewers recommend 'Weak Accept', whereas one recommends 'Weak Reject'. I think the authors did a good job with their rebuttal, including new experiments and results which were requested by the reviewers. As such I recommend acceptance pending the condition that the authors include these new results in their camera-ready version.

**Paper Type:**

validation/application paper

**Special Issue:**

no

---

### Decision · Program_Chairs · 2020-04-11

Accept